# Features of the Degradation of the Proton-Conducting Polymer Nafion in Highly Porous Electrodes of PEM Fuel Cells

**DOI:** 10.3390/membranes13030342

**Published:** 2023-03-16

**Authors:** Andrey A. Nechitailov, Polina Volovitch, Nadezhda V. Glebova, Anna Krasnova

**Affiliations:** 1Ioffe Institute, St. Petersburg 194021, Russia; 2Institut de Recherche de Chimie Paris (IRCP), Chimie ParisTech, PSL Research University, CNRS, 75005 Paris, France; 3Institut Photovoltaique Ile de France (IPVF), CNRS, Ecole Polytechnique, IPParis, Chimie ParisTech, PSL University, IPVF SAS, 91120 Palaiseau, France

**Keywords:** proton-exchange membrane, fuel cell, porous electrode, Nafion, carbon nanotubes, aging test, stability, degradation, membrane-electrode assembly

## Abstract

The stability of new membrane–electrode assemblies of a proton-exchange membrane fuel cell with highly porous electrodes and low Pt loading, based on the proton-conducting polymer Nafion, was characterized in conditions of electrochemical aging. A comprehensive study of the effect of the microstructure on the evolution of the electrochemical characteristics of the new assemblies was obtained by voltammetry, electrochemical impedance spectroscopy, X-ray powder diffraction, and scanning electron microscopy. Because high (>70%) porosity provides intensive mass transfer inside an electrode, structural-modifying additives—long carbon nanotubes—were introduced into the new electrodes. PEM fuel cells with electrodes of a conventional composition without carbon nanotubes were used for comparison. The aging of the samples was carried out according to the standard accelerated method in accordance with the DOE (Department of Energy) protocols. The results show two fundamental differences between the degradation of highly porous electrodes and traditional ones: 1. in highly porous electrodes, the size of Pt nanoparticles increases to a lesser extent due to recrystallization; 2. a more intense “washout” of Nafion and an increase in ionic resistance occur in highly porous electrodes. Mechanisms of the evolution of the characteristics of structurally modified electrodes under electrochemical aging are proposed.

## 1. Introduction

Today, the issue of global energy transition and decarbonization is acute. Various technologies are used for the successful energy transition (e.g., wind energy and solar energy). However, the reliance of these resources on weather conditions and other shortcomings makes them unstable and intermittent during energy generation. These drawbacks have led to the need to store energy. Hydrogen energy storage is of interest because of the wide distribution of hydrogen in nature and its high energy density. Hydrogen can then be used as fuel in fuel cells (FCs). Proton exchange membrane (PEM) FCs are electrochemical devices that can efficiently convert chemical energy directly into electrical energy. PEM FCs possess notable advantages such as low operating temperatures, high efficiency, minimal maintenance, and compactness [1].

A well-known problem of modern PEM FCs is their insufficient lifetime. Several works were devoted to the issues of long-term PEM FCs and their individual components; see, e.g., [2,3,4,5,6]. The insufficient lifetime of PEM FCs affects their price and is a serious obstacle to their widespread adoption in the economy. In addition, insufficient operating times limit the range of the possible applications of these devices. Another serious challenge in PEM FC technology is to increase the electrochemical efficiency.

One of the ways to increase efficiency is to reduce the mass transport losses in the electrode [7]. The high porosity of the electrode makes this possible, thus increasing the power density [8,9]. For newly developed materials, the permanence of such highly porous electrodes over time under prolonged electrochemical exposure is questionable and needs to be studied.

As is well known, PEM FC electrodes in the classical form are composite structures; they have mixed conductivity and consist of a catalyst on a carbon carrier and an ionomer. In the case of FCs with a PEM, Nafion is usually used as an ionomer, and Pt is used as a catalyst. Due to the pronounced catalytic properties of Pt during the operation of the device, there might be a strong effect on the corrosion of the electrode components, primarily on the corrosion of the carrier (as a rule, the carrier is carbon black), and it could itself be subjected to the corrosive effects of oxygen, water, and other components in contact with its surface in one way or another.

Nafion is a typical perfluorinated sulfonic acid material (membrane or dispersion) with a hydrophobic polytetrafluoroethylene (PTFE) backbone and side chains terminated with hydrophilic sulfonic acid groups [10]. The PTFE backing must provide excellent chemical and mechanical stability. At the same time, Nafion is prone to colloidization and migration in the electrode.

The issue of the stability of electrochemical energy conversion systems has been topical for many years [11,12,13] due to several circumstances. Firstly, electrochemical systems (electrolyzers, fuel cells, supercapacitors, etc.) are inherently corrosive with respect to their components, and second, their cost strongly depends on the service life of the devices, which in many cases limits the commercialization of the product. Moreover, the development and continuous improvement of such devices, aimed at increasing efficiency and productivity, is associated with the use of new materials, the design of new electrode structures, and other deviations from traditional technological solutions, and this, in turn, is associated with the need for appropriate studies of the longevity of such systems.

A large number of publications are devoted to the degradation of membrane–electrode assemblies (MEAs) of PEM FCs, and in particular, a separate group of them focuses on the degradation of individual forms of nanostructured carbon [14,15]. The use of materials such as carbon nanotubes (CNTs) and graphene-like materials in the composition of electrodes offers several advantages in terms of increasing both the electrode activity [16] and the intensification of mass transport processes due to the formation of a special kind of structure [17]. The improved characteristics of new electrodes containing CNTs and graphene-like materials motivated us to study their long-term operation and degradation mechanisms during operation.

CNTs are used in PEM FCs to improve some properties of the electrodes [18,19,20]:
−Improve mass transport by increasing the porosity and the number of large transport pores [19];−Increase the performance of PEM FC through cocatalysis [20];−CNTs have corrosion resistance and thermal stability, and they are used as catalyst carriers with a high specific surface area.

In [18], it was noted that CNTs have numerous superior properties, including electrical conductivity, thermal conductivity, mechanical strength, and the ability to support catalysts by providing an increased surface area. CNTs offer great promise for overcoming the problems of existing FCs. In [19], it was noted that a CNT sheet as a functional interlayer would be advantageous in improving the performance of commercialized PEM FCs.

At the same time, an obvious problem with the use of CNTs is the stability of the electrochemical system during long-term operation.

To study the resistance of MEA to electrochemical exposure in laboratory conditions, the USA Department of Energy (DOE) developed a system of accelerated stress tests (ASTs), which allow for the assessment of the stability of one or another part of a system in a relatively short time. To assess the longevity of a catalyst, multiple superpositions of triangular voltage pulses in the range of 0.6–1.0 V are used. These ASTs are often used in catalyst longevity studies.

This approach, based on multiple cyclic applications of the potential under certain conditions to assess the long-term viability of a catalyst, has been used in many works (e.g., [21,22]).

In [21], single triangular potential scans with different upper and lower limits were applied to FCs containing electrodes with different Pt/C compositions to evaluate the catalytic effect of Pt on the corrosion of a carbon carrier with a large surface area. The rate of carbon loss in H_2_/N_2_ and air/air modes was determined by integrating the resulting peaks of CO_2_ concentration in the exhaust gases of the positive electrode. In the work [22], a single-element PEMFC was used to study the degradation of an industrial membrane electrode catalyzed by Pt by conducting AST with a rectangular voltage in the potential range from 0.6 to 1.0 V.

An analysis of the scientific literature shows that practically no attention has been paid to the issues of ionomer migration during the functioning of the MEA. In addition, we note that most of the works are aimed at stabilizing the membrane itself, and the problems of Nafion stability in the electrode are practically ignored.

In accordance with this, the goal of the present work was to study the features of the degradation processes of proton-conducting components of PEM FCs with highly porous electrodes and low Pt loading containing CNTs in comparison with traditional systems without CNTs to find out which degradation processes are critical for these systems.

## 2. Experimental Section

### 2.1. Materials

Samples of MEAs with two types of electrodes were studied:Traditional electrodes containing platinized carbon black and Nafion proton-conducting polymers;Structurally modified electrodes containing the addition of CNTs.

Table 1 presents the composition of the studied samples of MEA.

Dispersions of solid components in a water–alcohol medium of the corresponding component composition served as the initial electrode material for these samples.

MEAs were fabricated using composite electrode materials containing platinized carbon black (Pt/C), a structure-modifying additive: multi-walled CNTs and a Nafion-type proton-conducting polymer, as described below.

Platinized carbon black (Pt/C).

A commercial E-TEK (The Fuel Cell Store, Bryan, TX, USA) product containing 40% Pt on Vulcan XC-72 carbon black was used [23]. The specific surface area was 200 [24]–250 m^2^/g [25,26,27,28].

### 2.2. Taunit MD CNTs

Multiwalled CNTs of the Taunit MD type (LLC NanoTechCenter, Tambov, Russia) were used. The characteristic structural properties of these nanotubes were large length-to-diameter ratios amounting to 670–2500 and high porosity due to pores being larger than 100 nm [29]. CNTs were purified from the catalyst additives used during their fabrication by acid treatment, washed with distilled water, and dried. Nitric acid diluted with water in a volume ratio of 1:1 was used. The treatment was performed at a temperature of about 100 °C for 15 min.

### 2.3. Proton-Exchange Polymer of Nafion Type

The commercial product, i.e., a 2-propanol–water solution of Nafion with a concentration of 20% (DE2020, Ion Power Inc., DuPont, Wilmington, DE, USA), was used. The ionomer was introduced into the electrode material. MEAs were prepared on an MF4-SK-type PEM (OJSC «Plastpolymer», St. Petersburg, Russia) that is an analog of the Nafion 212-type membrane (see Table 2).

### 2.4. Preparation of Dispersion

The preparation of the electrode material dispersion included two stages: the mechanical and ultrasonic dispersion of the mixture of components in the 2-propanol–water solution. The volume ratios of the 2-propanol–water liquid components were in the range of 1:1–1:5. The ratios of the solid to liquid phases in the final dispersion were in the range of 1:40–1:80. A sample with CNTs was prepared using a method assuming the preliminary coagulation of Nafion from its solution in the liquid phase, followed by its introduction into the electrode structure [31,32]. For coagulation, a commercial Nafion dispersion with the required concentration was diluted with water in a volume ratio of 1:1 prior to its addition to the dispersion.

Mechanical dispersion was performed in a Milaform MM-5M magnetic stirrer with a rotation speed of about 400 rpm (the magnetic stirring bar was covered with Teflon) to obtain a visually homogeneous mixture (without visible blobs) (~0.5 h). Subsequent ultrasonic dispersion was carried out in a Branson 3510 ultrasonic bath for 40–100 h until a homogeneous dispersion that did not delaminate within 1 min was obtained.

### 2.5. MEA Preparation

MEAs were prepared by depositing a homogeneous dispersion of components directly on a PEM through a stainless-steel mask. Prior to the deposition of the electrode material, the membrane was kept for 15 min in 0.5 M sulfuric acid at a temperature of 70–80 °C and then washed five times with water. The electrodes were prepared by depositing dispersion components in the 2-propanol–water solution directly on a Nafion-type PEM (MF4-SK) with a thickness of 50 μm. The deposition was carried out sequentially, first on one side of the membrane, then on another. To achieve this, a 5 × 5 cm^2^ membrane was placed on a stainless-steel substrate 6 × 6 cm^2^, 2 mm thick, which was placed on the surface of the IKA C-MAG HP 7 wafer with a heat controller. A 5 × 5 cm^2^ stainless-steel mask was placed on the membrane surface, in the center of which there was a 1 × 1 cm^2^ window. After thermostatting the system at a temperature of *T* = 85 °C for ~15 min, an electrode layer was applied, layer by layer, with each layer being dried. The amount of applied material was controlled gravimetrically after cooling the membrane for ~15 min in an air-dry state. The amount of the deposited material was monitored gravimetrically.

Before performing electrochemical measurements, MEAs were kept in 0.5 M sulfuric acid for 15 min at a temperature of *T* = 70–80 °C and were then washed five times with distilled water.

### 2.6. Characterization and Aging Procedures

The morphology of the carbon materials was studied by the low-temperature nitrogen adsorption method on the ASAP 2020 analyzer by Micromeritics (USA). The specific sample surface area was calculated using the BET method, volume, size, and size distribution of pores were calculated using the DFT method. The bulk density and specific pore volume were measured gravimetrically.

Raman spectra were recorded with a Renishaw Confocal Raman Microscope (UK) using a green laser (532 nm, 100 mW power). For the Nafion study, 15 µL of a 10% Nafion dispersion was applied to a silicon plate and then covered with glass. MEAs were investigated as they were.

The testing of electrocatalysts in the MEA was carried out as follows:

The MEA was placed in a standard measuring cell (FC-05-02, ElectroChem Inc., Woburn, MA, USA) with graphite current collectors with the following characteristics: temperature maintenance that ranged from room temperature to 180 °C, gas overpressure *P* = 0–2 atm., and electronic resistance less than 10 mΩ [33]. Toray 060 standard carbon paper was used as the gas diffusion layer.

MEAs were subjected to electrochemical aging according to the accelerated DOE method at room temperature and atmospheric pressure for a given number of cycles (0, 100, 300, 1000, etc.) of a voltage scan in the range of *E* = 0.6–1.0 V at a sweep rate of *v* = 50 mV/s. Wet (≈100%) N_2_ and H_2_ were applied to the electrodes. An aging electrode was supplied with N_2_. The MEA aging regime corresponded to that of the DOE protocols (Table A-1 in [34]).

During the electrochemical action, the change in the MEA characteristics caused by aging was monitored by the current–voltage characteristic (CVC) method at a sweep rate of *v* = 10 mV/s. The two-electrode method was used with the P-150 potentiostat (LLC Electrochemical Instruments, Chernogolovka, Russia). Before the measurements, the MEA was activated as described in [35]. The area of the electrochemically active surface area (ESA) of Pt was periodically measured by cyclic voltammetry under N_2_/H_2_ flow (by hydrogen desorption).

The electrochemical impedance spectra (EIS) were recorded on the Z-500X instrument with the AX-500PL attachment (LLC Electrochemical Instruments, Chernogolovka, Russia) in the H_2_/N_2_ humid gas flow (relative humidity close to 100%).

The hodographs were recorded at a voltage close to the open circuit voltage (polarization 0–100 mV) with an alternating voltage amplitude of *A* = 8 mV and a charge transfer resistance of *R_CT_* > 80 Ω in the frequency range of *f* = 500 kHz–0.1 Hz. The parasitic inductance of the cell was taken into account and corrected in the high-frequency region of the hodographs using the ZView2 program [36].

X-ray diffraction (XRD) patterns of initial and aged samples were collected on the X’Pert X-ray diffractometer (Malvern Panalytical Ltd., Malvern, UK) with Cu Kα radiation (λ = 1.54060 nm). The step size was 0.1313028°, and the time per step was 599.25 s. The diffraction patterns were processed using the HighScore Plus 3.0.5 program. The Rietveld method was used to refine and determine the crystallite size and microstrains.

### 2.7. Calculations of Electrochemical Characteristics from the Experimental Data

The ESA of Pt was calculated from the ratio:*S_Pt_* = *Q_des_*/210(1)
where *S_Pt_*—Pt ESA, cm^2^; *Q_des_*—charge spent on hydrogen desorption, µC; 210—coefficient relating the charge to the surface, µC/cm^2^.

The Pt loading was gravimetrically calculated by the ratio:*G_Pt_* = *M_CL_* ∗ *N* ∗ 0.4(2)
where *G_Pt_*—Pt loading in the electrode, mg/cm^2^; *M_CL_*—catalytic layer weight, mg; *N*—proportion of Pt/C in the catalytic layer; 0.4—proportion of Pt in E-TEK.

The specific mass catalytic activity (CA) of Pt in the oxygen reduction reaction (ORR) was calculated from the known relation:CA = *J@E*/*G_Pt_*(3)
where CA—Pt CA at a potential of *E* = 0.9 V vs. NHE, A/mg (Pt); *J@E*—current density of ORR at a potential of *E* = 0.9 V vs. NHE, A/cm^2^; *G_Pt_*—Pt loading in the electrode, mg/cm^2^.

The Pt specific activity (SA) in ORR was calculated by the ratio:SA = (*J@E* ∗ *S_CL_*)/*S_Pt_*(4)
where SA—Pt SA at a potential of *E* = 0.9 V vs. NHE, mA/cm^2^ (ESA Pt); *J@E*—current density ORR at a potential of *E* = 0.9 V vs. NHE, mA/cm^2^; *S_CL_*—visible electrode surface area, *S_CL_* = 1 cm^2^; *S_Pt_*—Pt ESA, cm^2^.

## 3. Results and Discussion

### 3.1. Structural Characteristics

Table 3 lists some of the structural characteristics of carbon materials: the platinum support (Vulcan XC-72 carbon black) and the structure modifier (CNTs).

Figure 1 compares the pore size distribution for the two materials.

It can be seen from the data presented in Table 3 and Figure 1 that the specific pore volume of CNTs is much larger than that of carbon black at comparable values of the specific surface area. This is due to structural features of CNTs that have a high length/diameter ratio and form a framework with large pores.

Figure 1 shows that CNTs also have higher porosity compared to carbon black in the pore range up to 100 nm.

Figure 2 shows the Raman spectra of the pristine MEA and the MEA sample after aging. It can be seen from the figures that the structure of the Nafion spectra (wavenumber range ~1150–100 cm^−1^) for these two samples is qualitatively similar, and there is a triad with a strong central band at 916 cm^−1^. At the same time, the relative intensity (with respect to carbon lines D and G) of the Nafion lines in the case of an aged sample is lower. This fact may indicate various causes of degradation and requires further research. The reasons may be, for instance, the washing out of a part of Nafion from the electrode, or its partial destruction with the washing out of more mobile reaction products.

### 3.2. Electrochemical Characteristics

The CVCs of the studied samples in the process of aging are shown in Figure 3. For both the traditional samples (type 2) and the samples containing CNTs (type 1) in the composition of the electrodes, an abrupt deterioration in electrochemical characteristics after a certain number of cycles is observed. The main feature seen from the figure is a change in the slope of the middle linear part of the CVC, corresponding to an increase in electrical resistance and the disappearance of the “diffusion bend” of the final part of the CVC. Since the ionic resistance in such systems is much higher than the electronic resistance, this indicates that the ionic conductivity is significantly degraded during aging and is responsible for losses limiting the electrochemical process.

Cyclic voltammograms were taken to compare the surface area of Pt before and after AST (Figure 4). The surface area of Pt was estimated from the charge spent on the desorption of hydrogen from the surface of Pt.

It can be seen from Figure 4 that the cyclic voltammograms during aging change differently for different samples and different electrodes. For the H_2_ electrode with CNTs, the polarization capacitance of the electrical double layer increased significantly after aging, as evidenced by the relatively high current density. The surface area of Pt for electrodes with CNTs decreased to a lesser extent than for electrodes of traditional composition, both for H_2_ and N_2_ electrodes.

To assess the degradation of the electrocatalytic activity of the electrode, the Pt ESA was measured (Figure 5) and the Pt CA in the ORR was calculated as the current density at *E* = 0.9 V divided by the mass of Pt in the electrode (Figure 6). Under such conditions, with small polarization, the electrochemical process proceeded in the region of the reaction kinetics control. There were therefore practically no diffusion losses, and the result was determined by electrocatalytic activity. Note that the Pt CA in the ORR for the sample with CNTs was significantly (~2 times) higher throughout the entire aging process, which is consistent with the results of our work [20]. At the same time, the Pt SA for the initial sample with CNTs was ~1.5 times higher (see Table 1, Figure 5 and Figure 6). Figure 5 and Figure 6 show that both parameters (ESA and CA) for both samples changed symbiotically: they gradually decreased. The decrease in the CA for 10,000 cycles for both samples occurred by ~1.8 times. The situation was somewhat different for ESA. The decrease in the Pt ESA over 10,000 aging cycles was less for the sample without CNTs than for the sample with CNTs: the decrease was 1.3 and 1.8 times, respectively. This indicates that the Pt SA for the sample without CNTs decreased by ~30% during aging, while the Pt SA for the sample with CNTs remained at the same (initial) level. Characteristic data are given for samples of two types.

EIS values of the studied samples are shown in Figure 7. From the analysis of the high-frequency region responsible for the ionic resistance of the electrode [36,37,38], it follows that during the degradation of the samples, the ionic resistance increased and the slope of the region responsible for the ionic resistance decreased. This was associated with the appearance of the inhomogeneity of the electrode resistance [35,39].

Quantitative data on the change in ionic resistance, calculated from the EIS data, are summarized in Table 4. For the sample with CNTs (type 1 MEA), a jump in resistance precisely at 3000 cycles of electrochemical exposure was observed. For the sample without CNTs (type 2 MEA), the resistance grew more smoothly and reached 760 mΩ after only 5000 cycles. These results are in good agreement with the CVC dynamics (Figure 3). We also note that the decrease in the slope (tg(a), Figure 7) of the linear section of the EIS for the sample without CNTs occurred more smoothly and by a smaller amount (up to 0.8 versus 0.5 for type 1 MEA).

### 3.3. Microstructure Evolution

The crystal structures of Pt/C catalysts were examined by XRD. XRD patterns for initial Pt/C, Pt/C, and Pt/C+CNTs after aging electrocatalysts are shown in Figure 8. For all cases, the diffraction peaks at about 40°, 46°, 68°, 81°, 86°, 104°, 118°, and 123° were due to Pt (111), (200), (220), (311), (222), (400), (331), and (420) reflections, respectively, which confirmed a face-centered cubic structure of Pt (ICDD 03-065-2868). This is in good agreement with the literature [40,41].

The intensive peak at 2θ = 26.8° for the Pt/C+CNTs sample, corresponded to graphite, plane (002) [42]. For other samples, although it was noticeable, it was less pronounced, which could be explained by the absence of highly ordered forms of graphite in samples not containing CNTs.

Table 5 shows the structural characteristics of Pt particles for the samples of different compositions and with different prehistories obtained from the XRD spectra.

Several important conclusions can be made from Table 5.

The Pt unit cell parameter varied, and in all cases, it was less than the usual value (3.923 Å—Standard JCPDS card no. 00-004-0802). The initial Pt/C samples with or w/o CNT had the smallest cell parameter. With aging, the unit cell parameter increased, and to a greater extent for the sample w/o CNTs (the value reached 3.917(2) Å, while for the sample with CNTs, it was only 3.911(5) Å). This could be attributed to two possible effects: 1. the size effect [43] (the influence of the carrier substrate [44,45] on the cell parameters) and 2. Pt recrystallization during aging. During Pt recrystallization under relatively stationary conditions during aging, the crystals tend to form a cell without distortion. Following this logic, we can say that in the sample w/o CNTs, Pt underwent recrystallization during aging to a greater extent than in the sample with CNTs.

This was confirmed by a change in the coherent scattering region (CSR), which is usually associated with the size of crystallites, and a change in micro stresses. For the initial sample and the aged sample with CNTs, these parameters had fairly close values. The CSR for the sample with CNTs slightly increased after aging, while the shape of the crystallites (the CSR values for different crystallographic directions) did not undergo significant distortions, and the micro strain also practically did not change, while for the aged sample of the traditional composition, the micro strain of platinum crystallites decreased, the CSR increased, and the crystallites acquired a more uniform shape along the crystallographic directions.

Some SEM observations of the surfaces before and after aging are illustrated in Figure 9. Pt agglomerates of larger size seemed to be visible on the top surface of the Pt/C sample w/o CNTs aged for 10,000 cycles in H_2_/N_2_ gas flow (brilliant zone in Figure 9b). So, large agglomerates were not evidenced on the surfaces of the Pt/C with and w/o CNTs aged in the same conditions (Figure 9a,c).

All this indicates a significantly greater Pt change during aging in a sample that did not contain CNTs. Since the processes of Pt recrystallization under electrochemical exposition occur with the participation of the PEM Nafion, it can be argued that in the sample with CNTs, due to the spatial mismatch, Pt contacted Nafion to a lesser extent and was more stable during aging.

In addition, it cannot be ruled out that, as a result of the recrystallization of Pt in samples without CNTs, its SA decreased due to the size effect but also due to a decrease in micro strain (see the above discussion of Figure 5 and Figure 6). However, this requires additional research.

An analysis of the obtained results suggests that the degradation of highly porous electrodes in the presence of CNTs occurs somewhat differently than electrodes of traditional composition and structure. In the case of highly porous electrodes (with CNTs), a significant contribution is made by an increase in the resistance to proton transfer in the electrode due to its migration and loss of contact with Pt nanoparticles. In this case, the decrease in the area of the electrochemically active surface of platinum in electrodes with CNTs occurs to a large extent due to the loss of this contact with Nafion, that is, due to the loss of part of the Pt from the electrochemical process, whereas in traditional electrodes, Pt recrystallizes to a greater extent and its particles increase in size, which lead to a decrease in the surface area.

The idea of losing some of the contact between Pt and Nafion is consistent with the EIS data (see Figure 7). The decrease in tg(a) during aging reflects the appearance of inhomogeneity in the proton conductivity in the electrode [35,39] associated with Nafion migration.

## 4. Conclusions

Higher stability of the electrocatalytic activity characteristics of Pt, both CA and SA, was shown in the samples with CNTs compared to the conventional samples. A linear decrease in Pt ESA and a corresponding decrease in CA in terms of the initial Pt loading were observed during aging for both samples. While per 10,000 cycles for the sample with CNTs, a slight change in the Pt SA was observed, for the sample without CNTs, a decrease in the Pt SA by ~30% was observed. The latter was attributed to the recrystallization of Pt: an increase in the size of crystallites and relaxation of the micro strain. In the sample with CNTs, due to the spatial mismatch of the components, in particular, Pt and Nafion, the recrystallization of platinum was expected to be slowed down. As a result of the high porosity of the sample with CNTs, Nafion migration and a decrease in the homogeneity of the proton resistance of the electrodes were proposed to occur more intensively in it.

The degradation mechanism leading to the degradation of electrochemical performance upon aging for the sample with CNTs with high porosity was proposed to be due to Nafion migration in the MEA electric field through large pores, resulting in inhomogeneity and an increase in resistance to proton transfer. An increase in the resistance to proton transfer was also facilitated by the disruption of the interfacial region and the loss of part of the Pt from the electrode process due to the loss of contact with Nafion. In the sample with a traditional composition (without CNTs), these processes could also occur, but to a lesser extent. The decrease in the Pt ESA for the sample with CNTs seemed to be largely determined by the loss of contact with Nafion, while for the sample of traditional composition, this decrease was primarily attributed to recrystallization. These mechanisms need further verification by advanced chemical and microstructure characterizations of aged samples.

## Figures and Tables

**Figure 1 membranes-13-00342-f001:**
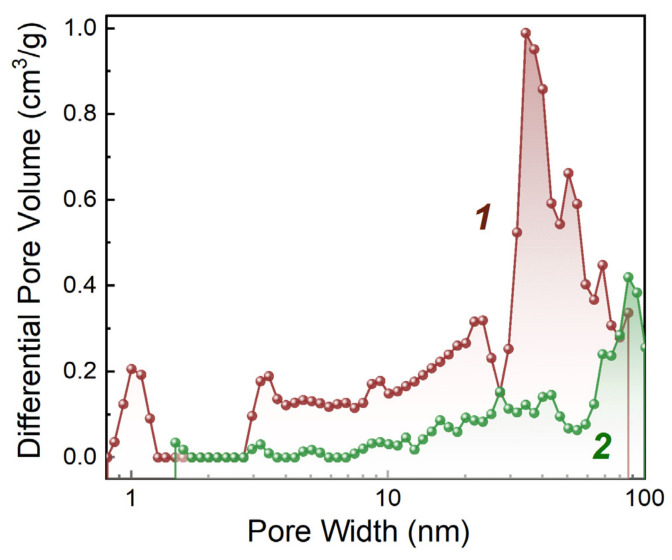
Pore volume distribution (according to DFT) in the range up to ~100 nm: 1—Taunit MD CNTs; 2—carbon black Vulcan XC-72.

**Figure 2 membranes-13-00342-f002:**
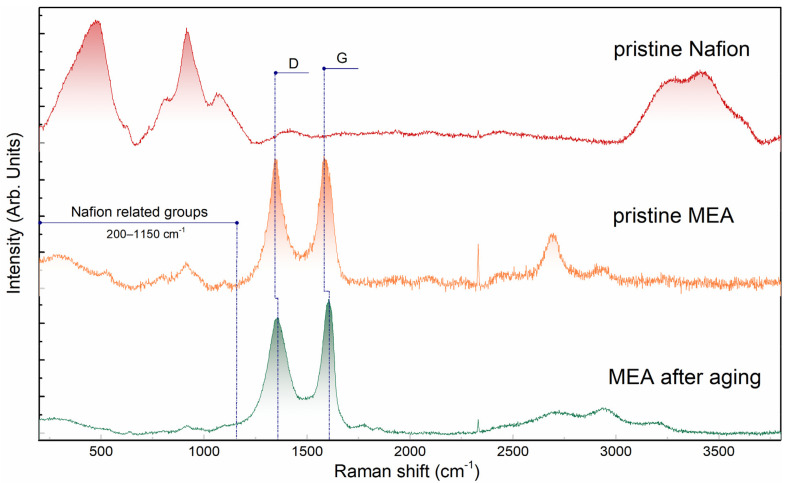
Raman spectra of Nafion dispersion (*C* = 10%) and membrane–electrode assemblies (type 1 MEA with CNTs) before and after 10,000 cycles of electrochemical aging.

**Figure 3 membranes-13-00342-f003:**
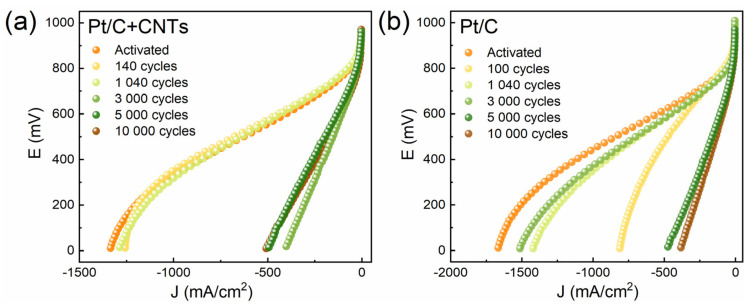
Change in CVCs during degradation of the cathode under N_2_ flow: (**a**) type 1 MEA (with CNTs); (**b**) type 2 MEA (w/o CNTs).

**Figure 4 membranes-13-00342-f004:**
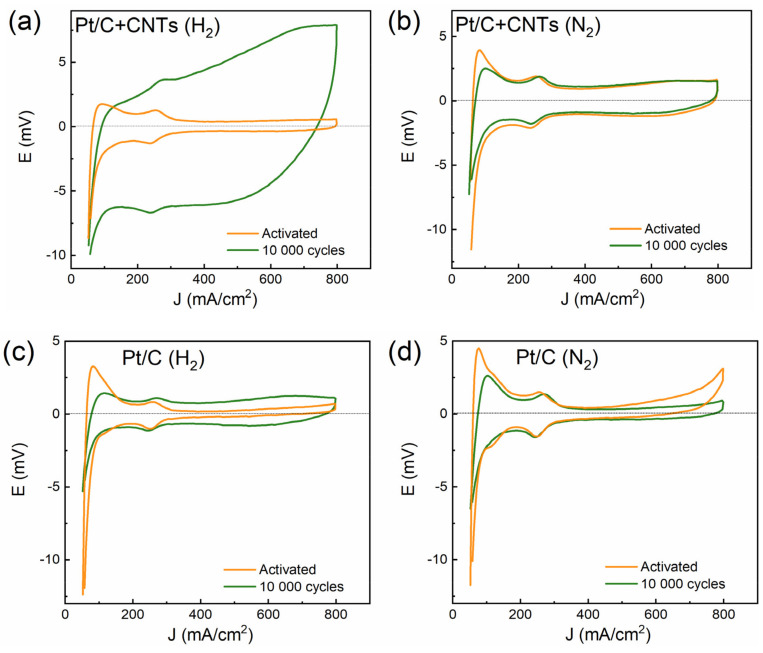
Cyclic voltammograms before and after 10,000 cycles of aging according to DOE stress test: (**a**) type 1 MEA (with CNTs), H_2_ electrode; (**b**) type 1 MEA (with CNTs), N_2_ electrode; (**c**) type 2 MEA (w/o CNTs), H_2_ electrode; (**d**) type 2 MEA (w/o CNTs), N_2_ electrode. Wet (≈100%) N_2_/H_2_; polarization rate 25 mV/s.

**Figure 5 membranes-13-00342-f005:**
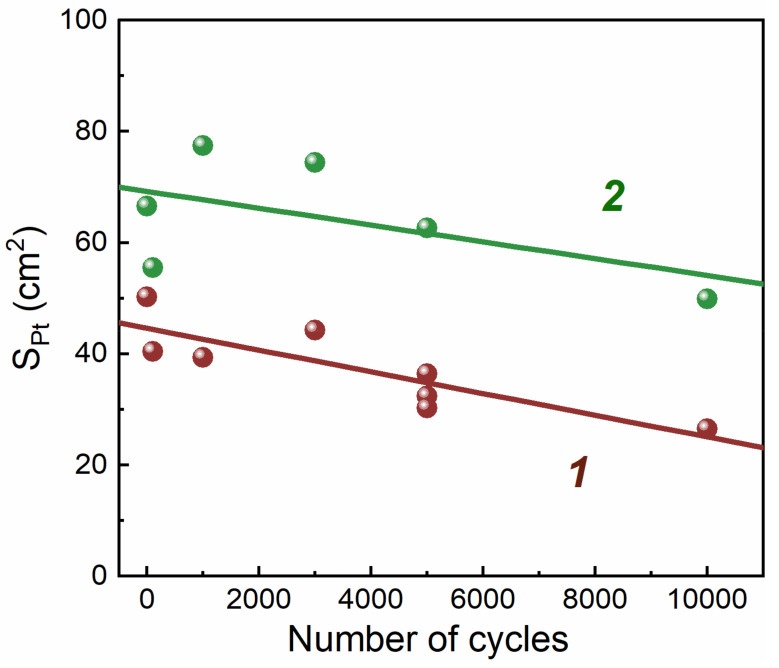
Change in the Pt electrochemically active surface area of the electrode during degradation (N_2_ electrode). 1—type 1 MEA (with CNTs); 2—type 2 MEA (w/o CNTs).

**Figure 6 membranes-13-00342-f006:**
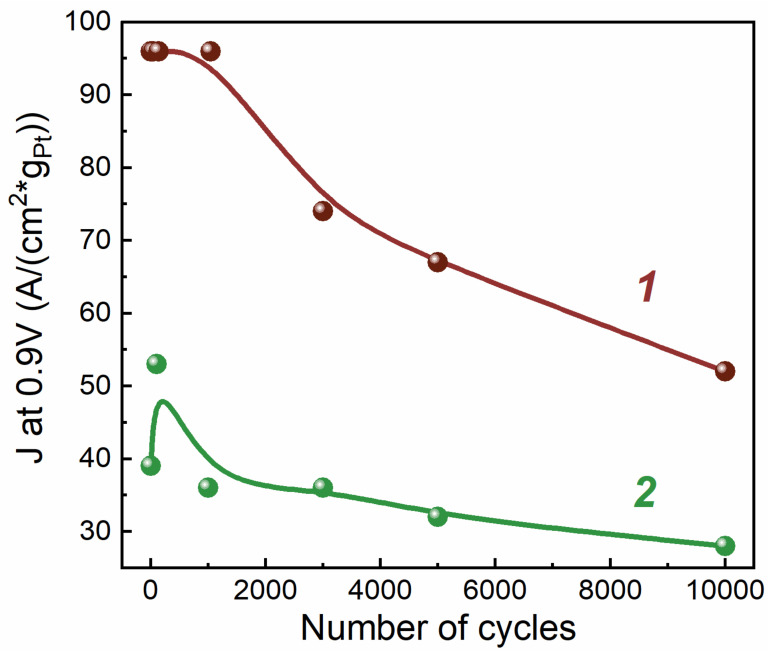
Change in catalytic activity at *E* = 0.9 V of MEAs during aging in terms of the initial Pt loading. 1—type 1 MEA (with CNTs); 2—type 2 MEA (w/o CNTs).

**Figure 7 membranes-13-00342-f007:**
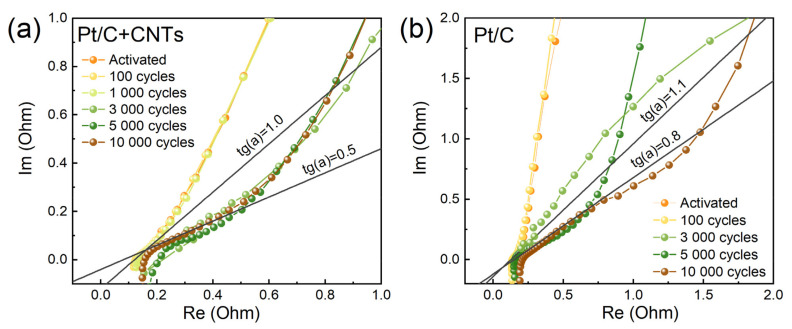
Evolution in the shape of the high-frequency section of EIS during degradation under N_2_/H_2_ flow: (**a**) type 1 MEA (with CNTs); (**b**) type 2 MEA (w/o CNTs).

**Figure 8 membranes-13-00342-f008:**
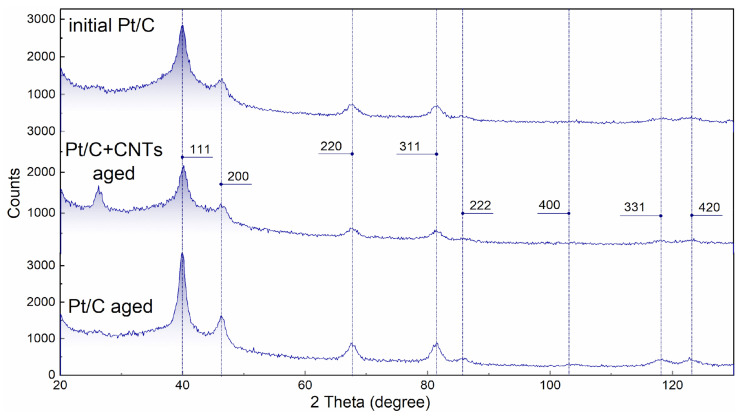
X-ray diffraction patterns of initial Pt/C electrodes of traditional composition, Pt/C+CNT electrodes after aging, and Pt/C electrodes of traditional composition after aging.

**Figure 9 membranes-13-00342-f009:**
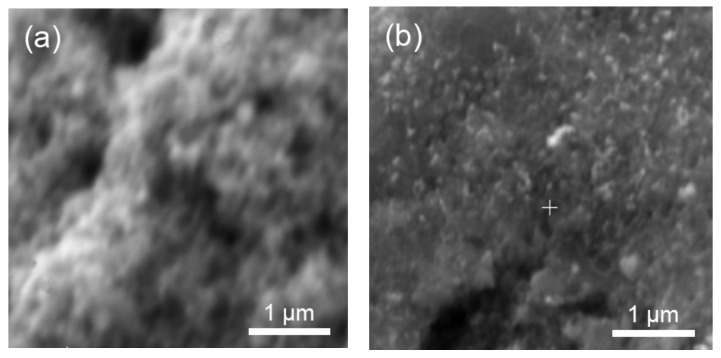
Secondary electron images of the electrodes: (**a**) initial Pt/C w/o CNTs, (**b**) Pt/C w/o CNTs after 10,000 cycles of aging in H_2_/N_2_ gas flow, (**c**) Pt/C with CNTs after 10,000 cycles of aging in H_2_/N_2_ gas flow.

**Table 1 membranes-13-00342-t001:** Composition of samples used for degradation study.

Samples	Composition	Cathode Pt Loading, mg	Electrode Porosity, %
Type 1	Pt/C(40%Pt) + CNTs 30% + coagulated Nafion 40%	0.12	74.7
Type 2	Pt/C(40%Pt) + noncoagulated Nafion 20%	0.25	59.5

**Table 2 membranes-13-00342-t002:** Some characteristics of the MF4-SK membrane (data from OJSC “Plastpolymer”) in comparison with the Nafion 212 membrane [30].

Membrane Type	Thickness, μm	EW	Chemical Stability	Ω at RH~100%, Ohm*cm	Available Acid Capacity, meq/g
MF4-SK	45 ± 5	1000	chemically stable	8	1.0
Nafion 212	50.8	1100	chemically stable	10 (measured by EIS)	0.92 minimum

**Table 3 membranes-13-00342-t003:** Some structural characteristics of carbon materials.

Material	Bulk Density, g/cm^3^	Porosity, %	Specific Pore Volume (Including Macropores), cm^3^/g	Specific Surface Area, m^3^/g
Vulcan XC-72 (Pt carrier)	0.26	88	1.7	200–250
CNTs Taunit MD	0.025–0.060	99–97	40–16	≥270

**Table 4 membranes-13-00342-t004:** Change in the ionic resistance of the cathode during degradation, mOhm.

Number of Cycles	Sample of Type 1	Sample of Type 2
initial (100)	120 ± 4	100 ± 3
1000	120 ± 4	80 ± 2
3000	900 ± 27	300 ± 9
5000	1000 ± 30	760 ± 23
10,000	1000 ± 30	1300 ± 39

**Table 5 membranes-13-00342-t005:** Some crystallographic characteristics of Pt in samples.

Sample	Average Crystallite Size, nm/ Micro Strain for Pt(hkl), %					Cubic Unit Cell a = b = c, Å
	(111)	(200)	(220)	(311)	(222)	
Pt/C w/o and with CNTs initial state	2.03/ 5.5	2.04/ 4.8	1.66/ 4.2	1.48/ 4.0	1.46/ 3.9	3.909(3)
Pt/C with CNTs after 10,000 aging cycles	2.20/ 5.2	2.17/ 4.6	1.76/ 4.0	1.56/ 4.0	1.54/ 3.9	3.911(5)
Pt/C w/o CNTs after 10,000 aging cycles	2.88/ 3.9	3.04/ 3.2	2.73/ 2.5	2.40/ 2.5	2.37/ 2.4	3.917(2)

## Data Availability

The data presented in this study are available on request from the corresponding author.

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
