# Peer review of "Features of the Degradation of the Proton-Conducting Polymer Nafion in Highly Porous Electrodes of PEM Fuel Cells"

_membranes, 2023, doi:10.3390/membranes13030342_

Round 1

Reviewer 1 Report

1. The length of the abstract is more than 200 words, and it needs to be refined. Abstract: The abstract should be a total of about 200 words maximum [see https://www.mdpi.com/journal/membranes/instructions].

2. The number of keywords is too small to represent the main content of this article. At least, 'fuel cell/PEMFC',  'Nafion', 'highly porous electrodes', 'carbon nanotube' and so on should be included.

3. Page 2, Due to the pronounced catalytic properties of Pt during the operation of the device, it might have a strong effect on the corrosion of the electrode components, primarily on the corrosion of the carrier (as a rule, it is carbon black), and could itself be subjected to the corrosive effects of oxygen, water, and other components in one way or another in contact with its surface. 

    This is the first time I have seen that Pt metal will cause corrosion. I think it should be the degradation of membrane material that causes corrosion because of the formation of hydrofluoric acid.

4. Page 3, In addition, we note that the main part of the work is aimed at stabilizing the membrane itself, and the problems of Nafion stability in the electrode are practically ignored.

    What causes the instability of Nafion? What is the structure of Nafion? Which functional groups cause instability? It should be briefly introduced. Nafion is a typical perfluorinated sulfonic acid membrane with hydrophobic PTFE backbone and side chains terminated with hydrophilic sulfonic acid groups [10.1002/er.4875]. The PTFE backbone should provide excellent chemical and mechanical stability.

5. Page 4, Does the preparation of MEA adopt hot-pressing technology? I think the details of the current MEA preparation process are still insufficient.

     Temperature maintenance range from room temperature to 180°C, will it lead to the degradation of the Nafion membrane?

6. The membrane degradation should be related to the chemical structure change in the Nafion polymer, hence, FTIR and Raman spectra may make more sense. Now the SEM and XRD, absorption, and all the characterizations are related to catalysts or the electrode parts, not membranes at all. The title is Features of the degradation of the proton-conducting polymer Nafion, but for Nafion, characterizations are very limited. See [10.1021/ja2074642; 10.1016/j.memsci.2020.118517]

Reviewer 2 Report

1. This study focuses on the catalyst layer of the MEA, which is less related to the concept of “proton-conducting membranes,” the research topic of this special issue. If the authors could describe more about the relationship between the catalyst and electrolyte membrane, that would be better.

2. The unit “mkm” in SEM image is a Russian term meaning micrometer. If that is the case, please use the symbol μm to replace it. 
